# Development and Analysis of a Distributed Leak Detection and Localisation System for Crude Oil Pipelines

**DOI:** 10.3390/s23094298

**Published:** 2023-04-26

**Authors:** Safuriyawu Ahmed, Frédéric Le Mouël, Nicolas Stouls, Gislain Lipeme Kouyi

**Affiliations:** 1Univ Lyon, INSA Lyon, Inria, CITI, EA3720, 69621 Villeurbanne, France; 2Univ Lyon, INSA Lyon, DEEP, EA7429, 69621 Villeurbanne, France

**Keywords:** WSN, IoT-based monitoring systems, distributed systems, distributed leakage detection, distributed leakage localisation, crude oil pipelines

## Abstract

Crude oil leakages and spills (OLS) are some of the problems attributed to pipeline failures in the oil and gas industry’s midstream sector. Consequently, they are monitored via several leakage detection and localisation techniques (LDTs) comprising classical methods and, recently, Internet of Things (IoT)-based systems via wireless sensor networks (WSNs). Although the latter techniques are proven to be more efficient, they are susceptible to other types of failures such as high false alarms or single point of failure (SPOF) due to their centralised implementations. Therefore, in this work, we present a hybrid distributed leakage detection and localisation technique (HyDiLLEch), which combines multiple classical LDTs. The technique is implemented in two versions, a single-hop and a double-hop version. The evaluation of the results is based on the resilience to SPOFs, the accuracy of detection and localisation, and communication efficiency. The results obtained from the placement strategy and the distributed spatial data correlation include increased sensitivity to leakage detection and localisation and the elimination of the SPOF related to the centralised LDTs by increasing the number of node-detecting and localising (NDL) leakages to four and six in the single-hop and double-hop versions, respectively. In addition, the accuracy of leakages is improved from 0 to 32 m in nodes that were physically close to the leakage points while keeping the communication overhead minimal.

## 1. Introduction

Pipeline transportation of crude oil is increasingly being adopted because of its low carbon footprint compared to other modes of transportation, such as ships, waterborne vessels, and rail trucks, and it is also considered the safest means of transportation [1,2]. However, it sometimes fails as a result of third-party interferences, ageing infrastructure, corrosion, erosion, equipment failures, material and weld failures, natural hazards, and operational failures [3]. These failures lead to the release of hydrocarbon in the environment into the form of oil leakages and spills (OLS) causing environmental pollution and degradation, loss of life and revenue, health-related issues, and socioeconomic impacts [1,2,4,5]. Consequently, various leakage detection and monitoring systems (LDMSs) are being adopted to reduce the impacts of such failures, such as wired and wireless systems, statistical detection and localisation, hybrid systems, robots, and remote sensing systems.

However, incidents in pipeline transportation continue to occur. For instance, Shell Nigeria recorded a loss of crude oil in their network to the tune of 11,000 barrels per day in 2018 [6], an increment of approximately 550% in comparison to the previous year. Additionally, some of the classical leakage detection techniques (LDTs) that comprise community-based surveillance and security personnel, supervisory control and data acquisition (SCADA) systems, fibre optics, conducting cables, helicopters, and satellites are difficult to maintain, inflexible, have limited coverage, high detection and localisation (DAL) times, high false alarms, or are expensive to implement [7,8,9].

Hence, in recent years, industries have seen an increase in the adoption of WSN and IoT-based systems for infrastructure monitoring. The oil and gas industry covers three sectors (upstream, midstream, and downstream) to improve the various processes related to each sector. The midstream sector is used to monitor the various failures encountered in the transportation of oil and gas products. While research studies [2,10,11,12,13] have shown that it is more efficient compared to other LDMSs, some challenges still exist. For instance, energy consumption, reliability, the accuracy of leakage DAL, the sensitivity of detection communication and node failures, and single point of failure (SPOF) [14,15].

Therefore, in this work, we address these issues by proposing a new distributed leakage DAL technique, HyDiLLEch, with the following contributions:A combination of existing LDTs to optimise their strengths and minimise their weaknesses while ensuring the continuous and accurate DAL of leakages.A presentation of the node placement strategy, which allows sensitivity in leakage DAL and elimination of SPOFs.An analysis of the efficiency of HyDiLLEch, considering the resilience to SPOFs, the accuracy of leakage DAL, the communication overhead, and energy consumption.

Moreover, this is an extension of the work by [16]. In this version, we also present the results of the detection sensitivity of different sizes of leakages, the accuracy of some classical LDTs, and a further representation of energy consumption. In addition, we provide an extended explanation of the design choices and specifications of the system. Finally, we elaborate on the algorithms to show the changes in sensing frequency according to the detection or localisation and change the metric of measurement for the accuracy to metres compared to the percentage in the first version, as in [17]. The rest of this article is structured as follows: In Section 2, we discuss the existing works; Section 3 details the hydraulic background of statistical leakage DAL. Section 4 discusses the main contributions of the work. Finally, we discuss our results in Section 5 and present our conclusion and future works in Section 6.

## 2. Related Works

Many LDTs and leakage detection monitoring systems (LDMSs) are used to reduce the negative impacts of OLS. In this section, we present research works on different aspects of designing an LDMS. In the first Section 2.1, we discuss several classical LDTs and LDMSs and present WSN/IoT-based monitoring systems in Section 2.2.

### 2.1. Pipeline Failure Detection and Localisation Techniques

Existing classical techniques for monitoring pipelines are based on research works and techniques, such as robots, computational fluid dynamics, and hybrid systems. Robots are mainly used to monitor pipeline integrity. You Na et al. [18] proposed the use of biometric robots for detecting anomalies in pipelines. With an insect-like crawling feature, the proposed robot can effectively travel along the different paths of the complex pipeline network. Sujatha et al. [19] proposed a prototype robot for real-time, continuous, and autonomous monitoring of the pipeline. This was enabled by integrating the robots with a mobile application for tracking changes, which could be customised to the needs. A preliminary test conducted on the usage of the robots shows the robustness and practicability of usage. Kim et al. [10] developed a sensor-based system for pipeline monitoring and maintenance by combining topology-aware robots, mobile sensors, and a radio frequency identification (RFID) sensor-based localisation technique. Their integrated system is expected to prevent failures through the early detection and reporting of anomalies, after which, the robotic agents provide recovery by conducting repairs. Several experiments were conducted showing the cost-effectiveness and scalability of their system. However, research on robotics for overall pipeline monitoring is still in its early stages and has not been largely adopted due to its implementation complexity and the performance limitations for industrial purposes. Currently, pipeline inspection gauges are more commonly used to monitor structural integration with various limitations, such as one-directional movements.

Unlike robots, non-intrusive techniques exist, such as those based on computational fluid dynamics to detect and localise leakages in pipelines. Ostapkowicz, in his work [14], demonstrated the use of statistical leakage detection techniques based on pressure gradients and negative pressure waves. Experimental results were obtained by taking the pressure measurements and the fluid transmission speed. Both LDTs had good localisation accuracy and differed mainly in terms of energy consumption, i.e., the negative pressure wave had higher energy consumption because of its sampling rate compared to the gradient-based method, with lower energy consumption and accuracy. Similarly, Beush-ausen et al. [20] proposed transient leakage detection through an integrated analysis of the pressure and flow rate and the use of modified volume balance. With this technique, leakage detection was shown; however, some challenges persist. For instance, the localisation error was up to ten kilometres, some discrepancies in the flow metre were recorded, and there were limitations on the communication bandwidth and the effect of the type of fluid on this result. On the other hand, Santos et al. [21] proposed a detection method based on the usage of Doppler ultrasonic flow metres and transit-time ultrasonic flow metres. They carried out simulations to determine the impact of air bubbles on the accuracy of the detection and to understand its effectiveness. Karray et al. [22] also presented a leak detection technique based on the modified time of arrival difference, predictive Kalman filter, and the system-on-a-chip wireless sensor node. However, this work focused on reducing the energy consumption of LDMSs by incorporating data filtering, preprocessing, and compression in the algorithm. While existing statistical techniques present high accuracy in the DAL of leakages, challenges still exist, including energy consumption and single point of failure (SPOF) related to centralised implementation.

Other classical methods of DAL exist based on remote sensing, such as satellites, helicopters, vehicles, and unmanned aerial vehicles (UAVs) using radar, RGB images, or specialised cameras. S, Kostianoy et al. [23,24,25] presented the efficiency of using satellite systems for monitoring various pipeline transportation aspects. These include differentiating anthropogenic and natural effects of pipelines, ports, terminals, and the ecological impacts on the water bodies in proximity. The environmental impacts were evaluated using an integrated sea track web model to measure the spatial and temporal characteristics. Likewise, other works [26,27] demonstrated the use of a hyperspectral satellite for remote sensing in pipeline transportation by monitoring chemicals, gases, and other dynamic changes on the land, enabling the early detection of pipeline degradation. Although satellite-based monitoring provides multi-dimensional monitoring for pipelines and is suitable for difficult terrain, it is expensive to implement. Human-based monitoring systems, on the other hand, involve the use of observer vehicles, helicopters, or UAVs for the visual DAL of leakages (in addition to remote sensing, in some cases). Security personnel and community-based surveillance are also used to detect leakages, specifically those resulting from vandalism because of the size. However, studies [7,8,9] show that these techniques are expensive, ineffective, and impractical due to the very long detection times (days to weeks) and hazards to the environment.

As discussed in Section 1, IoT-based monitoring systems are increasingly being adopted for infrastructural monitoring in various industries, including the oil and gas industry. Thus, in the following subsection, we discuss various research works on these systems.

### 2.2. WSN/IoT-Based Pipeline Monitoring

IoT-based systems are enabled by wireless sensor networks (WSNs) consisting of multiple sensors for data collection. Many research studies [12,28,29] have enumerated their advantages in terms of vandalisation, maintainability, and energy consumption, among others, for pipeline monitoring. Hence, in this subsection, we look at various aspects of the IoT ecosystem, such as architectural designs, placement strategies, LDT, and communication protocols, which play crucial roles in the efficiency of pipeline monitoring.

Khan et al. [7] proposed an IoT architecture consisting of three layers for the different sectors of the oil and gas industry. Their aim was to improve reliability and robustness through a hierarchical design that enabled performance enhancement and predictive maintenance via collaboration, reliable communication, and intelligent decision-making among the nodes. Yelmarthi et al. [30] proposed a four-layered architectural hybrid framework (wired and WSN). The aim was to provide a low-power and generic framework usable for diverse applications. Hence, various experiments involving damage detection, posture analysis, and physical activities were conducted to test the practicability.

In addition to architectural design, an important factor that affects the efficiency of the WSN is the sensor placement strategy. Berry et al. [31] proposed sensor placements in pipeline networks based on mixed integer programming (MIP), based on the temporal characteristics of the contamination impacts noted in the pipelines. Tests were carried out using EPANET, SNL-1, and SNL-2 to analyse the sensitivity of the placement to the contamination in a periodic manner. The sensitivity analysis was based on the consensus of the placement, i.e., where two sensor placements were defined as their intersection size divided by the number of sensors in each placement. The results show that the average pairwise consensus was in the range of 86.5% to 100%, with a maximum standard deviation of 1.3%.

Similarly, Sela et al. [32] worked on the efficient failure detection of water pipelines through optimal sensor placement based on multiple approaches. The preliminary step utilised an approximate solution of the minimum set cover of the minimum test cover (MTC) problem. They also proposed a novel technique using an augmented greedy MTC-based algorithm. To determine the efficiency of this approach, tests were conducted on a water network. The results show that the novel technique is three to eight times faster than the other. Lina et al. [33] extended these results by proposing a robust placement using a robust greedy approximation (RGA) and robust mixed integer approximation (RMIO) as enhancements to the nominal greedy approximation and mixed integer optimisation problem through redundancy. Experimental results show that both RGA and RMIO outperformed their nominal versions.

Likewise, Krause et al. [34] aimed to mitigate the intrusion of pipeline networks through robust sensor placements, they optimised this placement by utilising the minimax algorithm. The experimental results included the extension of the multi-criteria optimisation and efficient placement for large networks, i.e., up to 91% of the maximum placement score achievable. Sarrate et al. [35] studied the impact of sensor accuracy on infrastructural analysis using the isolability index, i.e., setting up the placement issue on a fault diagnosis-based performance maximisation criterion in contrast to the placement following non-linear physical laws leading to mesh networks. Tests conducted on water networks to detect leakages showed improvement in fault detection and the removal of the complexity of mesh connectivity in such extensive networks.

To reduce energy consumption and improve the sensors’ lifetime, Guo et al. [36] proposed a sensor placement on an oil pipeline by considering the maximum transmission range of the sensors and messages relayed in a hop-by-hop manner, which also reduced the number of sensors placed on the pipeline. However, Elnaggar et al. [37] took a different approach to the reduction of energy consumption through sensor placement for oil pipeline monitoring. His approach involved the use of ant colony optimisation and a genetic algorithm. The simulations, which were conducted on a pipeline segment to test the communication level, showed poorer performances in the genetic and greedy algorithms compared to the ant colony optimisation. However, they maintained similar results in the overall lifetime optimisation of the WSN. Al Baseer et al. [38] proposed sensor grouping and using an adaptive clustering algorithm for intermediate data delivery to reduce energy consumption. As a result, a significant reduction between 300% to over 500% was recorded as a result of the load-sharing mechanism among the cluster heads. Moreover, this approach resulted in up to a 62% reduction compared to the heuristic approaches. Further evaluation of the experimental results shows a 50% reduction in consumption. Li et al. [39], on the other hand, proposed a generic sensor placement to optimise energy consumption in a sensor network. To achieve this, they utilised retransmission and discrete power control for single- and double-tier uniformly and non-uniformly distributed WSNs.

Multiple LDTs have been proposed or implemented based on various WSN architectural designs. Sadeghioon et al. [40] proposed a novel algorithm for leakage DAL in underground pipelines aimed at improving the sensitivity of DAL. The algorithm is based on the relative measurement of the pressure and changes that occur as a result of the interaction between the hydrocarbon and the soil surrounding the area of the leakage. The test showed an improvement in sensitivity compared to the threshold-based techniques. They furthered their research [11] by introducing a specialised sensor based on a force-sensitive resistor for ultra-low-power DAL. Still on the reduction of energy consumption, Saeed et al. [41] proposed a reliable WSN-based system for oil and gas pipelines spanning over a large geographical area. The conducted preliminary tests on energy consumption showed promising results.

One of the key factors in designing an efficient and reliable WSN/IoT-based infrastructural monitoring system is the communication protocol utilised in the system. Sensor nodes used in WSN/IoT-based systems are constrained devices that are limited by several factors, such as energy consumption, which makes existing cellular communication networks, such as 3G, 4G, LTE, and others, unsuitable for their usage [42]. Hence, alternative protocols, as shown in Figure 1, are proposed to specifically cater to the needs of IoT systems. These protocols are broadly categorised into short- and long-range communication protocols and mainly differ in their topology, communication range, throughput, and energy consumption.

Short-range communication protocols include IEEE 802.15.4 (Zigbee), IEEE 802.15.1 (Bluetooth LE), 6LowPan, Wi-Fi, and others, with a maximum communication range of about 100 m. They share some similarities, such as a throughput of 20 kbps [43] and a mesh topology for Zigbee and 6Lowpan, while Bluetooth LE, on the other hand, has a point-to-point topology.

Long-range protocols (low power wide area network protocols—LPWANs) covering up to 50 km are further divided into licensed and unlicensed LPWANs [44]. The Third Generation Partnership Project (3GPP) NB-IoT is an example of a licensed LPWAN operating on a licensed spectrum of a 200k Hz band. Its connection density involves up to one hundred thousand devices. Sigfox and LoRaWAN, on the other hand, are examples of unlicensed LPWANs. They both operate on the 868 MHz band in Europe with LoRaWAN also operating in other radio bands. Sigfox uses an ultra-narrowband signal that further reduces energy consumption and is limited by the number of daily message deliveries depending on the data rate (usually up to 100 bps). LoRaWAN also has a low data rate, and the throughput depends on multiple factors, i.e., the size of the bandwidth and the spreading factor.

Thus far, we have presented works that considered the benefits of sensor placements, such as detection time, energy consumption, sensitivity, scalability, and robustness to failures, by proposing different strategies. However, various drawbacks exist; for example, the assumption that one sensor can detect failures in multiple pipelines, or their centralised implementations (both of which make them susceptible to SPOFs). Energy consumption and the accuracy of DAL are also other concerns in the industrial application of IoT-based monitoring systems. The communication protocol choice (to ensure coverage in the place of deployment) is also a factor to consider.

In the subsequent section, we present our contribution, HyDiLLEch. However, to discuss this, we present a background on fluid mechanics to justify our design choices.

## 3. Computational Fluid Mechanics

Fluid transmission is categorised into different flow types, each presenting distinct characteristics corresponding to changes, such as those in velocity or pressure over space and time. However, all fluid transmissions are subjected to the principle of fluid dynamics based on the conservation of energy, mass and linear momentum [45]. As such, the state equation (any data-driven equation) can be used to define the fluid phase behaviour [46,47] by measuring the fluid properties, such as temperature, flow, pressure, and density.

Crude oil transmission in a pipeline follows these principles and, therefore, presents certain measurable property changes (in both steady and unsteady states), i.e., with leakages and no leakages. Figure 2 shows the changes in the pressure gradient travelling in a pipeline from the inlet to the outlet, with the absence of leakages. In this figure, we can see a slope representation of the changes when it is presented in the time domain, where the inlet and outlet pressures (represented by $P_{0}$ and $P_{L}$, respectively) constantly decrease with distance at each measurement point. This representation differs significantly in the presence of the leakage as a result of the negative pressure wave (NPW) generated [14,48]. As shown in Figure 3, the generated NPW travels in the opposite direction from the leakage location, causes changes in the pressure at the inlet and outlet, and leads to the formation of two distinct gradients. $P_{0}^{leak}$ and $P_{L}^{leak}$ represent the inlet and outlet pressures when there is leakage. These properties (as well as other changes in the flow rates and temperature) are useful data in leakage detection, using statistical methods to advance the governing principle of fluid transmission, otherwise known as computational fluid dynamics (CFD).

CFD-based techniques include the real-time transient modelling (RTTM) mass volume balance technique (MVB), the pressure point analysis (PPA) technique, the gradient-based method (GM), and the negative pressure wave method (NPWM) [5,14,49]. In the following subsections, we present each of the aforementioned techniques including their advantages and disadvantages.

### 3.1. Real-Time Transient Model

RTTM is based on field instrumentation. It simulates pipeline monitoring through the hydraulic and thermodynamic properties by measuring the density, flow, pressure, temperature, and other properties of the fluid. These measurements are performed in real-time and represent the state information of the pipeline under all conditions. One of the important considerations in using RTTM involves deciding the boundary conditions, i.e., the input data, signal calculation, and processing [47]; this has a significant impact on the obtained results.

### 3.2. Mass Volume Balance

MVB technique is used to detect leakages when the mass balance at the outlet exceeds the threshold defined by Equation (Equation 1) [50]
(1)$$\epsilon \leq \rho_{in}-\rho_{out}-\frac{d}{dt}m_{p}$$
where ϵ is the defined threshold, $\rho_{in}$ is the fluid density at the inlet, $\rho_{out}$ is the fluid density at the outlet, $\frac{d}{dt}m_{p}$ is the change in pressure and temperature of the pipeline based on the liquid density and the cross-sectional area of the pipe.

The supervisory and data acquisition system (SCADA) typically uses the MVB in combination with other statistical methods, such as PPA for leakage detection and localisation in pipelines. The detection is determined using two variants of MVB, the *simple volume balance* and the *modified volume balance*. The simple volume balance is completely based on the principle of mass conservation, where both inlet and outlet masses are expected to be equal. The modified method, on the other hand, includes other state properties, such as the pressure or temperature to determine the leakage presence. The centralised nature of SCADA systems makes them susceptible to SPOFs. Additionally, they have high response times, are expensive, and are inflexible in regard to change [7].

### 3.3. Pressure Point Analysis (PPA)

The PPA technique is used to detect leakage by the periodic measurement of fluid pressure along the length of the pipeline. It uses the Bernoulli equation defined in Equation (Equation 2).
(2)$$z_{a}+\frac{P_{a}}{\rho g}+\frac{V_{a}^{2}}{2g}=z_{b}+\frac{P_{b}}{\rho g}+\frac{V_{b}^{2}}{2g}+E_{ab}$$
where $E_{ab}=\lambda \frac{V^{2}}{2gd}*L$ represents the energy head loss, λ is the coefficient of friction, *V* is the velocity of the fluid, *d* is the pipeline’s inner diameter, *L* is the distance between points *a* and *b*, $P_{a}$ represents the pressure at point *a*, ρ is the mass density of the fluid, *g* is the gravitational force, and $z_{a}$ is the elevation at point *a*.

To elaborate on the PPA technique, we consider the placement of several sensors along the pipeline, as shown in Figure 4. The measured pressure $P_{i}$ of each sensor is relative to its position on the pipeline. With the Bernoulli equation, this can be determined using the inlet pressure $P_{0}$, the energy head loss, the fluid velocity, and other parameters defined in Equation (Equation 2).

Thus, the statistical detection of leakage using PPA involves the evaluation of the determined pressure against a preset threshold. This non-computationally complex method makes the implementation of PPA easy. However, this technique cannot be used to localise leakage without combining it with other techniques (e.g., MVB) [51].

### 3.4. Gradient-Based Method (GM)

As discussed at the beginning of the section, leakage occurrence presents two distinct steady states, resulting in different PGs from the pipeline’s inlet to its outlet. These steady states, as shown in Figure 5, are before and after the leak location. The GM technique uses these changes in PG for DAL leakages.
(3)$$\bar{Q}=\frac{L\times dPG_{Q-L}^{leak}+\left(dp_{0}-dp_{L}\right)}{dPG_{Q-L}^{leak}-dPG_{0-Q}^{leak}}$$
where $\bar{Q}$ is the estimated leak location, *L* is the pipeline length, $dp_{0}$ is the average increment in the pipeline’s initial cross-section, $dp_{L}$ = average is the increment in the pipeline’s final cross-section, $dPG_{0-Q}^{leak}$ is the average increment in the pressure gradient before the point of leakage, $dPG_{Q-L}^{leak}$ is the average increment in the pressure gradient after the point of leakage.

For instance, we consider a leakage occurrence at a point *Q* in the pipeline, assuming that $PG_{a-b}$ denotes the pressure gradient between two points, *a* and point *b*, then $PG_{0-Q}^{leak}$ and $PG_{Q-L}^{leak}$ represent the two gradients before and after the leakage location. Note that $PG_{Q-L}^{leak}<PG_{0-L}<PG_{0-Q}^{leak}$ represent the differences in the two steady states used in GM DAL with Equation (Equation 3).

The GM technique is efficient in DAL with low computational complexity and energy consumption, and high accuracy.

### 3.5. Negative Pressure Wave Method (NPWM)

The negative pressure wave method (NPWM) is another LDT that is commonly used for DAL in pipelines [52]. Similar to changes in the steady state discussed in the previous subsection, an NPW is also generated in which the NPWM is based.

The generated wave, as shown in Figure 6, travels in the opposite direction from the leakage point with the speed of sound (*c*), calculated by Equation (Equation 4). Moreover, depending on the leakage size, the amplitude of the NPW attenuates as it travels along the pipeline. Hence, only sensors within the maximum detectable distance of the wave can detect the arrival of the NPW front. The formula for characterising the attenuation rate of the amplitude of the NPW signal is represented in Equation (Equation 5).
(4)$$c=\frac{1}{\sqrt{\rho (\frac{1}{K}+\frac{d}{Y.w})}}$$
where $A_{b}$ is the amplitude at sensing point *b*, $A_{a}$ is the amplitude at sensing point *a*, α is the attenuation coefficient, e=2.71 and *D* are the distances between two sensing points.
(5)$$A_{b}=A_{a}*e^{-\alpha D}$$
where $A_{b}$ is the amplitude at sensing point *b*, $A_{a}$ is the amplitude at sensing point *a*, α is the attenuation coefficient, e=2.71 and *D* are the distances between two sensing points.
(6)$$q=\frac{D-c\delta t}{2}$$
where *q* is the distance from the point of leakage to the nearest downstream node, *D* is the distance between the sensors, *c* is the negative wave speed, *t* is the communication time, and δt is the difference in the time of arrival of the signal in the upstream and downstream nodes.

Another important aspect of DAL is the time of arrival of the NPW front, which can be calculated using Equation (Equation 6) [53]. Both sensors at the sides of the leakage, i.e., the upstream and downstream sensors receive different times of arrival of the wave depending on their distances from the leakage. In general, NPWM has high accuracy in the DAL of leakages; however, due to its high sampling rate, the energy consumption is also high.

Given this background information, in the next section, we present our contributions.

## 4. Contributions

In this section, we present our contributions based on distributed leakage detection and localisation. As discussed in Section 1, the OGI in Nigeria experiences high rates of failure, such as vandalisation and operational failures. This work is aimed at providing an IoT-based monitoring system that can guarantee continuous and accurate detection and localisation of leakages in the presence of these failures. However, existing leakage detection techniques have various strengths and weaknesses. For instance, the PPA technique can be used to detect leakages but cannot be used to localise them and must be combined with other techniques. NPWM can be used for both detection and localisation but it has very high energy consumption, which is not practical for resource-constrained devices in an IoT-based system. Thus, we combined some of these techniques (details provided in the subsections) in a way that could optimise their strengths while minimising their weaknesses. Additionally, this was implemented in a distributed manner to remove the susceptibility to SPOFs experienced with classical LDTs. Globally, the objectives are as follows:Resilience: Remove the SPOF.Coverage and sensitivity: Allow optimal connections between the sensors and determine small-sized leaks.Accuracy: Ensure high accuracy in DAL.Energy efficiency: Minimise energy consumption without compromising accuracy.

In Section 4.1, we first discuss the system’s design and specifications covering different aspects. Then, we present and discuss the algorithm that was implemented on these design choices in Section 4.2.

### 4.1. Design Consideration and Specification

In our system of design and specification, we consider different aspects, as enumerated in the previous section. First, we define the system’s *architectural design* and the *communication protocol and network coverage* between the nodes and layers. Then we consider the *node placement strategy* for sensitive detection and resilience to failure.

#### 4.1.1. Architectural Design

Architectural designs for IoT systems are mostly application-based, ranging from centralised designs to distributed and hierarchical designs. In our system, we consider the efficiency of the system in regard to fault tolerance and communication aspects as the fundamental basis of our choices. Hence, we propose a three-layered hierarchical and distributed architecture aimed at reducing latency in the real-time DAL of leakages. In addition, the distributed architecture ensures robustness to SPOFs, communication failures, node failures, and other failures associated with third-party interference and natural occurrences in the pipelines.

Figure 7 shows the three-layered architectural design proposed, comprising *the sensor, fog, and cloud layers*.

At the sensor layer, we deploy sensors on the pipeline for the collection of information and data required for the implementation of services, such as data storage, data sharing, data preprocessing, detection, and localisation. The use of multi-spectral sensors, i.e., sensors with the capacity to collect diverse types of information, such as the negative pressure wave (NPW), its speed, and the pipeline pressure, enables this implementation. At the fog layer, we implement data and service management. The cloud layer, on the other hand, is reserved for long-term service implementations, such as historical data storage and predictive analysis. Next, we present the communication protocols proposed between the devices and the different layers.

#### 4.1.2. Communication and Network Coverage

In Section 2, we presented several communication protocols suitable for the constrained resources of IoT-based systems. Each protocol differs in throughput, communication range, latency, and energy consumption. In our work, we consider the protocols with communication ranges that can cover long transmission pipelines, covering hundreds to thousands of kilometres, their efficiency (in terms of energy consumption), scalability (when considered for industrial deployment) and, most importantly, their availability in the intended place of deployment (POD), which is Nigeria, in our case.

The LoRaWAN network communication protocol is, thus, particularly interesting for our use case. LoRaWAN is an example of an unlicensed LPWAN that is energy-efficient, cost-effective, easy to maintain and configure, and scalable. Its efficiency for long-distance communication (30 km and above) in out-of-line and rugged terrain was demonstrated in [54]. It is also fit for industrial purposes because a LoRa base station has up to 20 km of coverage with each cell having a connection capability of 50,000 devices. This provides a highly scalable network in comparison with its counterparts, such as NB-IoT [44]. In terms of deployment costs, a LoRa base station costs four times less than that of a Sigfox base station and fifteen times less than an NB-IoT base station [44].

To increase reliability in the communication aspect of the work, we also propose the use of an available cellular network, such as 3G, 4G, or LTE as a backhaul. Figure 8 presents a typical communication system architecture. The end devices communicate with each other via the LoRa communication protocol. Other short- or long-range communication protocols between the sensors and gateways could be used at the time of deployment as needed. Devices at the fog and cloud levels also exchange information with the cellular network as the backhaul between the fog and cloud layers. Next, we present a sensor placement strategy to ensure event coverage in the system.

#### 4.1.3. Sensor Placement for Event Coverage

In Section 2, we presented various sensor placement strategies to improve different aspects of LDMSs, including transmission capacity, the shortest distance between the node and event, or placements at key junctions, as shown in Figure 9 [32,33,36]. The drawbacks of these approaches include increased energy consumption when the nodes are always transmitting at their maximum capacity. Moreover, because of the long distance in the transmission lines, such placements fail to detect small leakages. Moreover, failures in intermediate nodes or nodes in key junctions can lead to interruption in the DAL of the leakage.

Hence, we propose the deployment of multiple sensors along each pipeline segment in the network, deploying several sensors along a pipeline segment in a network that is both robust to failure and sensitive to small leakages. We propose a new node placement strategy based on fluid propagation, specifically the NPW generated with the occurrence of the leakage in a pipeline.

Using the amplitude of the NPW (as shown in Figure 10) to determine the maximum detectable distance of small to big-sized leakages, we can ensure the sensitivity of detections by determining an optimal distance *D* on a pipeline of Lkm: For implementation, we apply the following constraints:

The distance (*D*) between nodes should be less than half of their maximum communication range of the sensor ($S_{cr}$) to ensure interconnectivity, data sharing, and resilience to SPOFs.
(7)$$D<S_{cr}/2$$The maximum detecting distance of the NPW ($NPW_{mdd}$, which will be experimentally determined) between nodes and the event source must be adequately small to guarantee sensitivity to small-sized leaks.
(8)$$0<D<NPW_{mdd}$$
where *D* is the distance obtained in Equation (Equation 7)With Equation (Equation 8), the distance (*D*) between the nodes, should be less than the $NPW_{mdd}$ between the upstream and downstream nodes surrounding a leakage, allowing the detectability of the NPW travelling in both directions.Finally, there should be at least three sensors from the total number of sensors deployed (*N*) that can detect the NPW front. Note that in an ideal case, both upstream and downstream nodes are enough to detect the arrival of the NPW front. However, to remove SPOFs as described at the beginning of this subsection, we add a redundancy of one node. This ensures the continuous DAL of leakages in the presence of node failure.
(9)$$\forall Q,[\exists \;n_{1},n_{2},n_{3}:n1\neq n2\wedge n2\neq n3\wedge n1\neq n3\;dist\left(n_{i},Q\right)<\left(NPW_{mdd}\right)]$$
where i>0 and *dist*($n_{i}$, *Q*) is the distance between node *i* and the leak location *Q*.

In the next subsection, we present our proposed distributed leakage detection techniques implemented in these design choices.

### 4.2. Hybrid Distributed Leakage Detection and Localisation Technique (HyDiLLEch)

HyDiLLEch is an LDT implemented in two versions (elaborated later) and based on the combination of some of the computational fluid dynamic detection techniques (i.e., PPA, GM, and NPWM) that were introduced in Section 3. Each LDT has strengths and weaknesses, such as leakage detectability, the accuracy of detection, and energy consumption. For instance, small leaks are detectable using the PPA LDT. It performs well under extreme conditions and has low maintenance. Both GM and NPWM can detect leakages in transient states [14]. Additionally, all three LDTs are non-invasive and can be easily deployed on existing infrastructures compared to the other LDTs [14].

However, several disadvantages are also associated with these LDTs. Consider the PPA technique, for example, it can neither detect leakages in a transient state nor independently localise [5]. The accuracy of DAL using the NPWM LDT relies on accurate detection of the arrival of the wavefront at the upstream and downstream nodes. The GM depends on the accuracy and calibration of the sensor nodes. As a result of these drawbacks, we propose the combination of these LDTs, i.e., NPWM, PPA, and GM, to take advantage of their strengths for the enhancement of DAL accuracy in a manner that minimises their negative impacts. Thus, we have HyDiLLEch, which is mainly aimed at increasing the *fault tolerance* of the LDMS, improving the *accuracy* in leakage detection, and minimising *false positives* in an *energy-efficient* and distributed manner, with the integration of multiple data sources.

The combination of these techniques enables their optimisation by clearly separating the detection and localisation phases and integrating other factors, as further explained in the following subsections.

#### 4.2.1. 3-Factor Leakage Detection

Leakage detection in HyDiLLEch utilises types of information, such as *predefined pressure threshold*, *pressure gradient*, and the *arrival* of the NPW front at the sensor. In this work, we consider a single horizontal pipeline, as shown in Figure 11 and DAL is based on this pipeline segment, as further explained below.

**Defining the pressure threshold:** HyDiLLEch begins by utilising the PPA LDT to pre-estimate the expected pressure at every node location in the pipeline. To determine the expected pressure, the elevation parameter (*z*) defined in Equation (Equation 2) is set to 0, with respect to the horizontal nature of the pipeline.Then, we calculate the PG to estimate the expected pressure at every sensor point by rewriting Equation (Equation 2) as follows:
(10)$$PG_{0-L}=\left(\frac{P_{0}}{\rho g}-\frac{P_{L}}{\rho g}\right)\frac{1}{L}$$
where $PG_{0-L}$ is the pressure gradient on a horizontal pipeline, i.e., elevation $z_{a-b}=0$ with length *L*, ($P_{0}$) is the inlet pressure, ($P_{L}$) is the outlet pressure, ρ is the fluid density, and *g* is the gravitational force.Once we determine the PG in a steady state, a threshold ϵ is set to accommodate the difference in the value read from the sensor and potential calibration error from the sensor readings. This threshold is set utilising the industrial permissible standard [14]. With this, the comparison between the actual sensor reading and the obtained value from the pre-estimation with Equation (Equation 10) is done. A difference greater than the threshold indicates possible leakage occurrence.**Difference in the pressure gradient:** The second step in the detection phase is to check the presence of the two PGs that must be present when leakage occurs, i.e., based on the fluid dynamic properties discussed in Section 3. Assuming a leakage point *Q* as shown in Figure 11, the two PGs are those formed between $n_{i}$ and $n_{i+1}$. Both PGs, i.e., $PG_{0-Q}^{leak}$ and $PG_{Q-L}^{leak}$ must be present and must respect the following conditions:
(11)$$PG_{0-Q}^{leak}\neq PG_{Q-L}^{leak}$$
(12)$$PG_{0-Q}^{leak}<PG_{0-L}$$
(13)$$PG_{0-L}<PG_{Q-L}^{leak}$$Ensuring the correctness of Equations (Equation 11)–(Equation 13) further lessens the high false alarms associated with other LDTs.**Arrival of the negative pressure wave:** As discussed in Section 3, all leakages must generate an NPW in the pipeline detectable by only a few sensors; the number varies based on the leak size. These sensors are noted as $NPW_{ds}$ in Figure 11 and are located around the areas where the pressure gradient is formed.The placements of these sensors (as we will see later in Section 5) are such that small leakages are detectable. Hence, the final step in the detection phase is to ensure the presence of the $NPW_{ds}$. In addition, for the arrival to be considered a positive detection, at least two (upstream and downstream) sensor nodes must detect the wavefront. This condition is centred on the waves travelling in opposite directions from a leakage point. Thus, we eliminate false positives by coupling the leakage detection to the arrival of the wavefront.Finally, the actual presence of a leak is confirmed following the enumerated factors. Then, the leakage localisation is estimated using a similar approach, discussed in the following subsection.

#### 4.2.2. Two-Factor Leakage Localisation

To localise the leakage, we combine the data (PG, NPW) of the sensors and narrow down the region of the leakage, i.e., the region where the NPW is detectable and the change of the gradient occurs. For instance, assuming a node $n_{i}$ as shown in Figure 11, which receives data from nodes $n_{i-1}$, $n_{i+1}$ and $n_{i-2}$, $n_{i+2}$ from the single- and double-hop neighbors, the leakage can be localised in two ways, as follows:(14)$${\bar{Q}}_{dpNPW}=\left(D\times i\right)+\left(D-q\right)$$
and
(15)$${\bar{Q}}_{dpGM}=\left(D\times i\right)+G$$
where ${\bar{Q}}_{dpNPW}$ is the leakage location based on the *partial information* on the NPW and estimated by Equation (Equation 6), ${\bar{Q}}_{dpGM}$ is the leakage location based on the *partial information* on the pressure gradient, and *G* is the distance based on the gradient calculation.

Equations (Equation 14) and (Equation 15) calculate the absolute leakage position; the localisation is first estimated when the first leakage, i.e., the front wave, is received. Based on the speed and amplitude of these waves, the leakage location can be refined using the pressure gradients, as described in Equation (Equation 3), or by analysing the arrival time of the NPW front:(16)$${\bar{Q}}_{dpNPW}=\left(D\times i\right)+\sum_{n=1}^{k}\left(c\times \delta t_{n}\right)÷k$$
where ${\bar{Q}}_{dpNPW}$ is an absolute leakage location, *n* is the *n*-th front end of the leakage wave, *c* is the negative wave speed estimated by Equation (Equation 4) of the *n*-th front end of the leakage wave, $\delta t_{n}$ is the difference in arrival time of the *n*-th front end of the leakage wave at the upstream and downstream nodes, and *k* is the number of received front-end waves.

We present the main structure of HyDiLLEch in Algorithms 1 and 2. As briefly mentioned in Section 4.1.2, it is implemented in two versions, depending on the neighborhood, i.e., 1-hop (2 nodes) or 2-hop (4 nodes); they are referred to as HyDiLLEch-1 and HyDiLLEch-2. Both algorithms have similar formulations, differing only in the number of nodes involved in the neighborhood collaboration.

They are also implemented on a single horizontal pipeline segment as shown in Figure 11. For the implementation of HyDiLLEch, the sensors used are multispectral, i.e., they can detect multiple fluid properties, such as pressure, speed, or temperature, and are equipped with their batteries.
**Algorithm 1** HyDiLLEch (Single-Hop)1:{Init-steady state}2:Set upper and lower PG thresholds {prevent noise}3:**for ever do**4:   {Activate nodes}5:   Obtain local $P_{i}$6:   **if** $P_{i}$ is greater than threshold **then**7:     Obtain pressure data from neighbours8:     Calculate local gradients $PG_{\left(i-1)-(i\right)}$ and $PG_{\left(i)-(i+1\right)}$ with neighbours9:     **if** PGs are outside the PG threshold **then**10:        Scan at high frequency11:        **for scanning time do**12:           Obtain $P_{i}$13:        **end for**14:        **if** exists $P_{i}$ greater than threshold, then share NPW data **then**15:          Localise using GM data (Equation (Equation 3))16:          Localise using NPW data (Equation (Equation 6))17:        **else**18:          No leak detected19:        **end if**20:     **else**21:        No leak detected22:     **end if**23:   **end if**24:   Sleep (duty cycle)25:**end for**

**Algorithm 2** HyDiLLEch (Double-Hop)
1:{Init-steady state}2:Set upper and lower PG thresholds {Prevent noise}3:
**for ever do**
4:   {Activate nodes}5:   Obtain local $P_{i}$6:   **if** $P_{i}$ is greater than P-threshold **then**7:     Obtain pressure data from neighbours8:     Calculate local gradients $PG_{\left(i-1)-(i\right)}$, $PG_{\left(i-2)-(i\right)}$, $PG_{\left(i)-(i+1\right)}$ and $PG_{\left(i)-(i+2\right)}$ with neighbours9:     **if** PGs are outside the PG threshold **then**10:        Scan at high frequency11:        **for scanning time do**12:          Obtain $P_{i}$13:        **end for**14:        **if** exists $P_{i}$ greater than threshold, then share NPW data **then**15:          Localise using GM data (Equation (Equation 3))16:          Localise using NPW data (Equation (Equation 6))17:        **else**18:          No leak detected19:        **end if**20:     **else**21:        No leak detected22:     **end if**23:   **end if**24:   Sleep (duty cycle)25:
**end for**



In the preliminary step on line 2 of Algorithm 1, the gradient thresholds are defined. The calculations are based on offline customisation to mitigate the classical variation of the fluid propagation model. On line 7 of Algorithm 1, each sensor captures the pressure at its respective location and propagates/receives the values to/from its predefined neighbour. The PG values are then computed to check for an unexpected pressure drop on line 8 of Algorithm 1. If so, several pressure values are sensed at high frequency—specified later—to accurately capture the wavefront of the leakage (lines 11–13 of Algorithm 1). If an NPW front is detected, the leakage localisation is computed using lines 15 and 16 of Algorithm 1. Afterward, sensors go back to sleep for a short period for the network duty cycle, i.e., the on-time and off-time periods.

In terms of the time complexity, let us consider the total number of activated nodes ($n_{activ}$) for each algorithm, the time taken to perform the arithmetic operations ($t_{arithm}$), the communication time between sensors ($t_{comm}$), and the scanning time ($t_{scan}$). As seen on line 7, $t_{comm}$ for every node is dependent on the number of neighbours considered, i.e., $n_{activ}-1$. Additionally, on line 10, there is a fixed time scanning frequency, and the number of times the localised pressure $P_{i}$ is retrieved can be represented as $n_{scan}$. Other operations, i.e., lines 8, 15, and 16 are arithmetic operations, which can be broken down (linear operations taking a unit of time to complete) into a total of $n_{arithm}$ operations. Thus, during the uptime of the duty cycle, the time complexity is $\left[n_{activ}\left(\left(n_{activ}-1\right)t_{comm}+\left(n_{scan}\right)t_{scan}+\left(n_{arithm}\right)t_{arithm}\right)\right]$. Further analysis shows that the time taken to complete operations involving the calculation, localisation, and sensing of the gradients is in the order of nanoseconds to milliseconds [14,55] and, thus, negligible. Hence, the most significant difference lies in the number of neighbours considered during the communication phase for each algorithm. For instance, the average latency for LoRa-based real-time communication is 7.6 s [56]. Accordingly, the average time complexity is $n_{activ}\left(n_{activ}-1\right)t_{comm}$. However, we enabled a time-out period, which is measured as the product of the round trip time and a constant coefficient to ensure packet delivery. In our case, we experimentally determine this coefficient to receive all packets as 1.3. Therefore, the worst-case complexity can be determined as $\left(n_{activ}^{2}\right)t_{comm}^{max}$, where $t_{comm}^{max}$ is the time-out period. Consequently, we can conclude that although the time complexity is in quadratic order, the total number of nodes activated for leakage localisation is a constant fraction of all of the nodes on the pipeline receiving the front wave (as we will discuss later in the results section). Therefore, the worst-case time is also constant. Hence, globally, both HyDiLLEch-1 and HyDiLLEch-2 can scale for larger deployments.

Furthermore, the distributed and localised sensor-based implementation of HyDiLLEch eliminates the SPOFs related to centralised systems. Additionally, it maintains high DAL accuracy with minimal energy consumption. The energy consumed is reduced by utilising only a fraction of nodes in the localisation process and low-frequency sensing during detection. Thus, we will discuss and analyse the simulation results in the next subsection.

## 5. Simulation Results

This work aims to determine the optimal distance between nodes for sensitivity to small leakages and remove SPOFs, while also efficiently improving the accuracy of DAL. The evaluation is based on a comparative analysis with the classical implementation of NPWM and GM in terms of susceptibility to SPOFs, the accuracy of DAL, and communication efficiency. For this implementation, we used NS3 to simulate crude oil propagation and the network aspect. Note that, to cover the typical distance of LoRA-based communications while implementing the neighbourhood connectivity among the nodes, we used the Wi-Fi protocol in NS3 with changes in some of its parameters, such as the transmission power. This is feasible since previous research [57] has demonstrated that the energy usage of Wi-Fi and LoRa varies by a constant factor once the distance reaches roughly 350 m in a linear setup similar to ours. Additionally, the properties of the pipeline and oil type (the material of the pipeline, oil velocity, and others) taken into consideration in this work were sourced from Nigeria’s Department of Petroleum Resources (DPR). We enumerate these properties and the simulation parameters in Table 1 and Table 2. In the following subsection, we discuss this sensor deployment on the pipeline.

### 5.1. Node Placement and Sensitivity to Small Leakages

This part of the work is focused on determining the optimal distance *D* between the sensors. Recall that in Section 4.2, one of the conditions in the detection phase was to enable sensitivity to small leakages. To achieve this, we analysed the impacts of various leakage sizes by using their amplitudes and pressure thresholds to determine their detectability through NPWM and PPA techniques. The simulation was conducted using small-, medium-, and large-sized leakages (0.5 kpsi, 5 kpsi, and 20 kpsi) at various distances between the sensors. In the obtained results, as shown in Figure 12, all of the sensors can detect the leakages for the PPA LDT using the pre-defined threshold value. Contrarily, the NPW has a maximum detectable distance of 2500 m for the leakage with the highest amplitude of 20 kpsi. Whereas, the $NPW_{mdd}$ for the least examined amplitude (0.5 kpsi) is about 1500 m.

Thus, we can conclude that the NPWM, i.e., the LDT based on NPW, is not as efficient as PPA LDT in the detection. However, the HyDiLLEch detection phase is also based on NPW, which must be detectable by multiple sensors. For NPWM, the number of sensors that detects the leakages decreases as the distance between the nodes increases with the smallest leakage having a guaranteed detection by the number of sensors required (Equation (Equation 9)) at a distance of 1000 m between the nodes. In addition, all of the other conditions defined in Section 4.2 are satisfied; hence, for this work, the optimal distance *D* is set to 1000 m.

With *D* determined, HyDiLLEch is implemented on a long-haul transmission pipeline for a single-phase laminar flow. The generated pressures in the simulation follow the Bernoulli equation, defined by Equation (Equation 2).

In this work, all simulations and tests were conducted in ideal conditions, i.e., no communication or node failures. The following subsections discuss the results.

### 5.2. Detection and Localisation Accuracy

The DAL tests were conducted using randomised leakage points drawn from the entire length of the pipeline. The confidence interval of these leakage points is between 9880 m and 14,523 m with a confidence level of 98%. In our previous work [16], we calculated the localisation accuracy based on a percentage, which has now been changed to distance in metres.

In the following subsections, we carry out a comparative analysis of two classical LDTs, i.e., NPWM and GM with HyDiLLEch.

#### 5.2.1. Classical Approach

The classical LDTs were implemented by collecting all sensed data at the centralised gateway and the obtained results are shown in Figure 13. The NPWM exhibits a high level of localisation accuracy, with an error of approximately 2 m in distance and little variance across all tested leakage locations. On the other hand, the GM has an average localisation accuracy error of approximately 227 m, with high variability in all of the tests conducted. The high accuracy obtained using NPWM can be attributed to the high sampling rate of all nodes, which results in higher energy consumption compared to GM, as we will see later.

While these results show the difference in the efficiency of both LDTs in terms of localisation accuracy, the centralised implementation, i.e., a single gateway for localising the leakages, makes them susceptible to SPOFs, which disrupts DAL. Thus, we will discuss how HyDiLLEch performs relative to these drawbacks, the corresponding accuracy in DAL, and the incurred overhead.

#### 5.2.2. HyDiLLEch

Unlike the classical implementation shown in the previous subsection, HyDiLLEch is implemented in a distributed manner. In this subsection, we discuss the results of the single-hop and double-hop versions separately. We also break down the results based on the number of nodes detecting and localising leakages (NDL) from the leakage point and the principal technique/information (dpNPW for NPW and dpGM for GM) used for the localisation. Moreover, the dpGM technique is activated on all nodes while the dpNPW is activated at a high frequency on the nodes whose gradient information differs from a neighbour, i.e., the nodes closest to the leakage area. Results for both versions are shown in Figure 14 and Figure 15. For the first comparison, in terms of NDL, the single-hop version shows an increment of three NDL compared to the centralised version with one NDL. This is made possible by the utilisation of the spatial data correlation (using each NDL geolocation, the pressure gradient, and the time of arrival of the NPW front) among multiple sensors surrounding the leakage point, represented as $n_{1–4}$ in Figure 14. Additionally, the average localisation errors from the actual leakage points for the best NDLs ($-n_{2}$ and $n_{3}$) are approximately 2 m and 33 m for dpNPW and dpGM, respectively. Note that these nodes are closest to the leakage locations. The remaining NDLs ($n_{1}$ and $n_{4}$), which are further away from the leakage points, have a higher average localisation error (298 m for dpNPW). In these nodes, localisation with dpGM is not possible as the extreme nodes are missing information from either an upstream or downstream node, which is not required for localisation with dpNPW.

Likewise, Figure 15 shows the results for HyDiLLEch-2 (the double-hop version) with a further increment in NDL, represented as $n_{1–6}$. The increased number of NDLs compared to HyDiLLEch-1 results from the wider collaborations in the nodes participating in the DAL process. Similar to HyDiLLEch-1, nodes that are closer to the leak locations ($n_{2}$, $n_{3}$, $n_{4}$, $n_{5}$) also present the highest and most improved accuracies. With dpNPW, the absolute localisation accuracy improved, with an average localisation error of 32 m across these NDLs. In addition, a lower variation was observed compared to HyDiLLEch-1. Furthermore, the average localisation at the extreme nodes ($n_{1}$ and $n_{6}$) is slightly reduced.
The results from the extreme nodes in both versions are kept potentially filterable at the fog layer. In the case of failures, the results from these nodes nonetheless give an average localisation error ranging between 230 and 367 m on the 20-km pipeline that we used. Furthermore, HyDiLLEch-1 and HyDiLLEch-2 showed significant improvements in NDL (up to 6) compared to 1 NDL for the centralised versions.

Finally, we note that the hybrid technique is loosely coupled to ensure the resilience of DAL in addition to enhancing the tolerance to SPOFs. This is particularly interesting in situations where performance-influencing factors, such as communication failure, node failure, and others, may be particularly detrimental to the information utilised for each localisation technique.

### 5.3. Communication Efficiency

The LDTs are analysed based on their communication efficiency by considering the communication costs, such as the number of packets, and the energy consumption. The obtained results are discussed in the following subsections.

#### 5.3.1. Communication Overhead

The total number of exchanged packets for the DAL process in each LDT is shown in Figure 16. In the case of the classical LDTs (NPWM and GM), this incorporates the information between the sensors and the gateways and represents the information shared among NDLs for HyDiLLEch-1 and HyDiLLEch-2. The results depict the highest number of exchanged packets for the NPWM, while the GM depicts the lowest number of exchanges. Both HyDiLLEch-1 and HyDiLLEch-2 have a communication overhead of 50% and 60%, respectively, compared to GM. However, this result is more efficient compared to the classical NPWM, with a complementary increment of NDL.

#### 5.3.2. Energy Consumption

The energy consumption evaluation for each LDT is based on the sampling rate used by each sensor as well as the radio energy consumption of the network duty cycle. Hence, the energy model is the sum of the energy consumption in different states, such as sensing, transmission, reception, sleep, and idle states based on the NS3 Wi-Fi energy model.

Figure 17 presents the energy consumption per sensor. NPWM, as shown in the figure, has the highest energy consumption because of its higher sampling compared to the other LDTs. The combination of data used by HyDiLLEch, including the separation of the detection and localisation phase (as discussed in Section 4.2), significantly decreases the energy consumption. With this, the sampling rate for the NDLs used in enhancing the localisation is less than 2% of the NPWM sampling rate and results in slightly better accuracy.

In Figure 18, we zoom into the energy consumption of GM and the two versions of HyDiLLEch to better understand the impact of the separation of the DAL phases. While GM still maintains a lower energy consumption, the overhead accrued by HyDiLLEch-1 and HyDiLLEch-2 is minimal considering the improvement in the localisation accuracy.

Analysis of the radio energy consumption rate of the network devices per duty cycle is also conducted. Figure 19 shows the most significant changes in energy consumption in the first two cycles, which is due to the varying connection requirements for the participating nodes in the compared LDT. Subsequently, the energy consumption converges in the cycles after the connections are established.

Figure 20 shows the cumulative energy consumption for each LDT. When compared to NPWM, the total energy consumed is reduced by 86% and 83% for HyDiLLEch-1 and HyDiLLEch-2, respectively, in the first cycle. However, an increment of 6–7% is observed as the overhead compared to GM, but the rate of increment is similar as the cycle increases.

## 6. Conclusions

In this work, we presented a systematic approach to the design, development, and analysis of an IoT-based system for monitoring crude oil pipelines. We also introduced a new LDT, i.e., HyDiLLEch, which is based on several classical LDTs (GM, PPA, and NPWM) and is implemented on a unique node placement strategy for the distributed detection and localisation of leakages. The optimal distance between the sensors based on the node fluid propagation-based node placement is experimentally determined as 1000 m. This distance ensures sensitivity and accurate detection of multi-sized leakages, especially small leakages with minimal cost overhead. HyDiLLEch successfully eliminates SPOFs associated with centralised systems by increasing the number of NDLs to four and six for the single and double-hop versions of the algorithm, respectively. In addition, the accuracy of DAL in the nodes closest to the leak point is improved by an average of 15 m (ranging between 0 and 32 m) with minimised energy consumption and communication overhead. While the energy consumption is considerably reduced compared to NPWM, there is still an overhead when both algorithms are compared to the GM. Hence, the first aspect of future work will be to implement data prioritisation at the sensor layer. This could further reduce energy consumption by sharing only highly prioritised data. In addition, while we have shown that HyDiLLEch removes SPOFs, we have not tested its performance in the presence of failures. Thus, the other aspect of future work will include the introduction of different kinds of failures, such as node failures, communication failures, and others in the system. The aim is to carry out a performance analysis of the system in the presence of these failures and to determine possible areas of improvement.

## Figures and Tables

**Figure 1 sensors-23-04298-f001:**
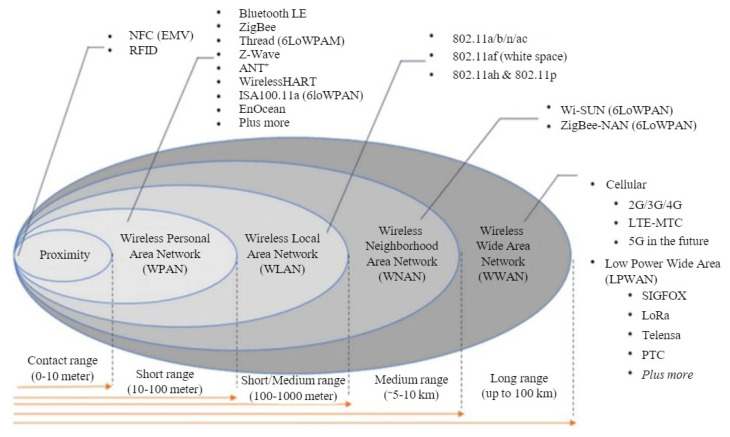
Wireless communication technologies [42].

**Figure 2 sensors-23-04298-f002:**
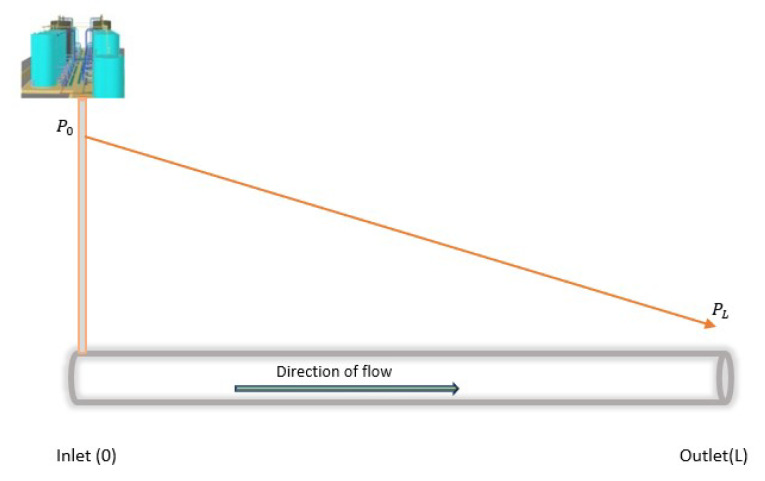
Steady fluid transmission in a pipeline.

**Figure 3 sensors-23-04298-f003:**
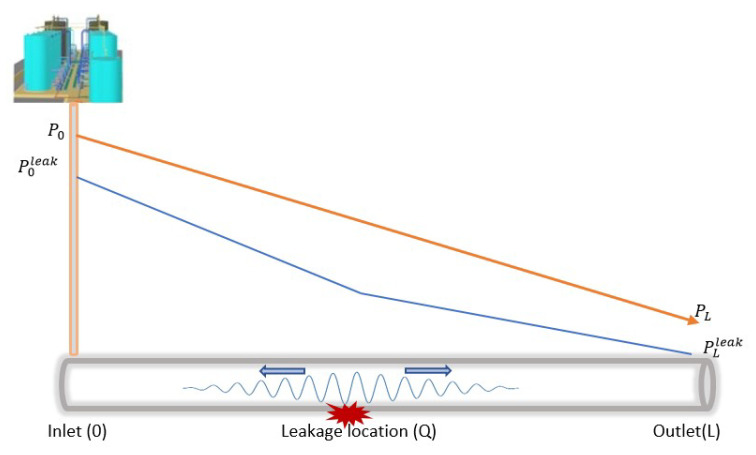
Leakage effects on the pressure gradient and NPW generation.

**Figure 4 sensors-23-04298-f004:**
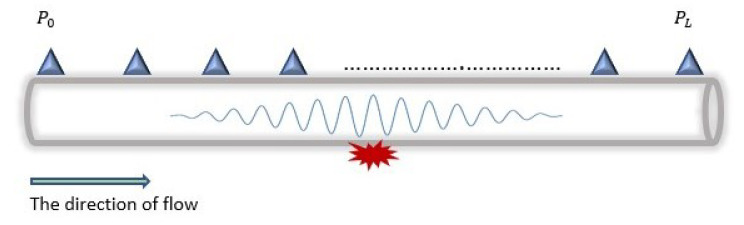
Pressure point measurements in a pipeline.

**Figure 5 sensors-23-04298-f005:**
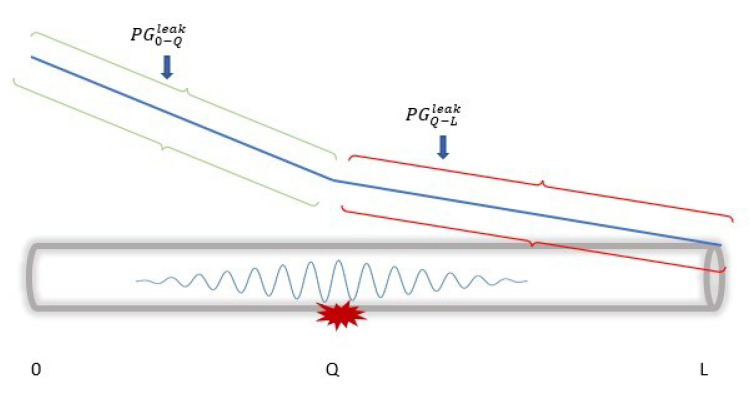
The pressure gradient distribution after a leak.

**Figure 6 sensors-23-04298-f006:**
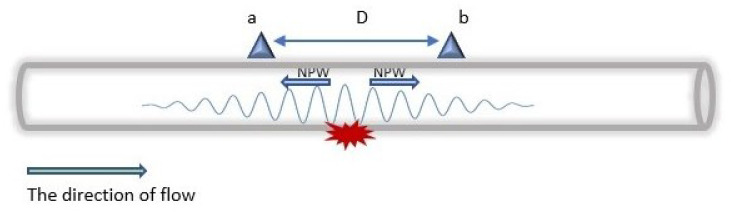
Negative pressure wave generated by a leak.

**Figure 7 sensors-23-04298-f007:**
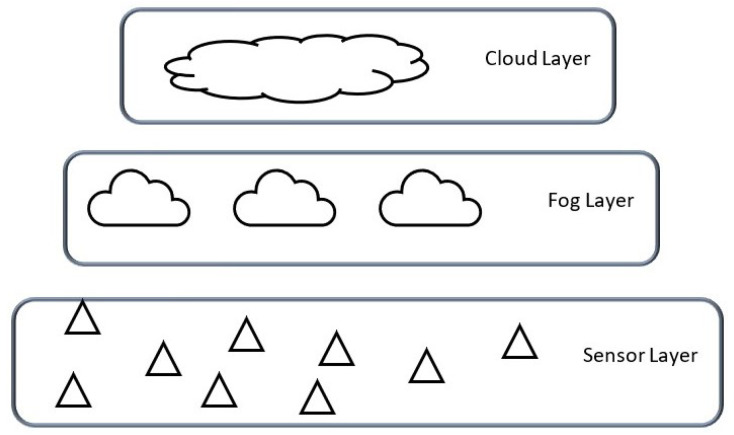
Network architecture.

**Figure 8 sensors-23-04298-f008:**
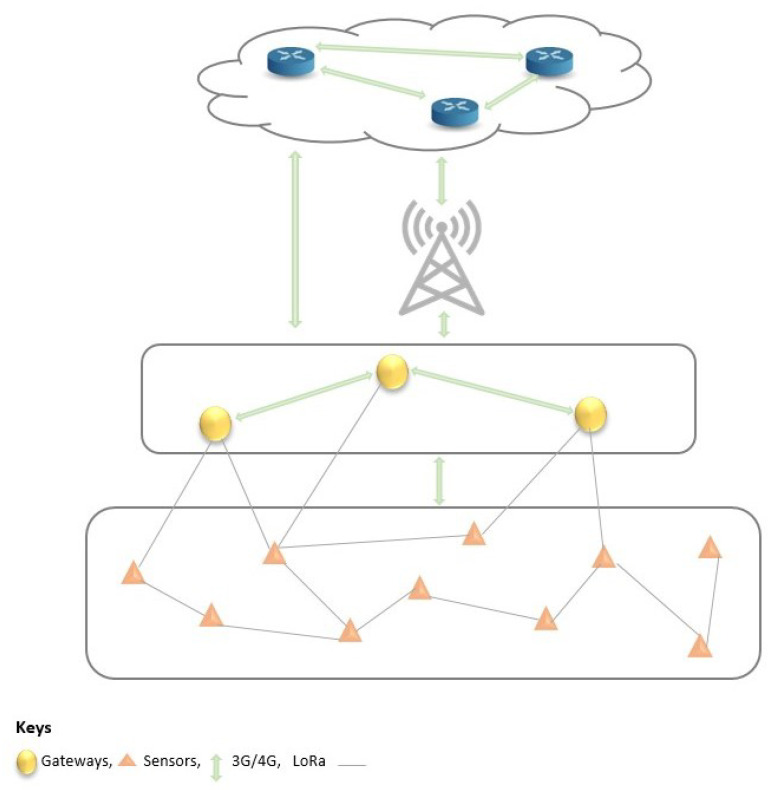
Network architecture with communication protocols.

**Figure 9 sensors-23-04298-f009:**
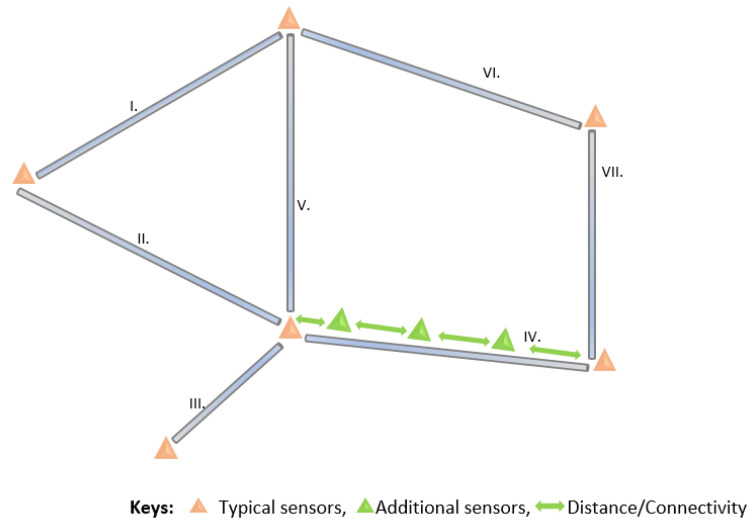
Sensor placement on long transmission crude oil pipelines.

**Figure 10 sensors-23-04298-f010:**
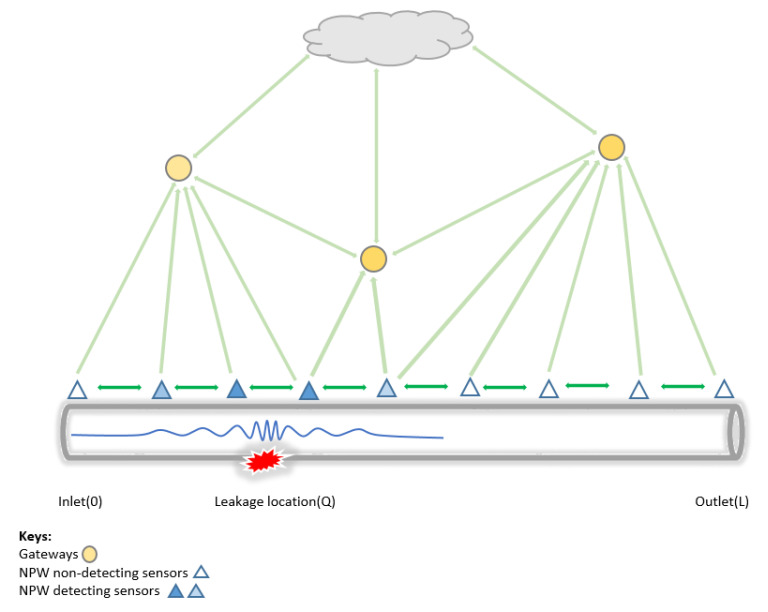
Event coverage on the proposed architecture.

**Figure 11 sensors-23-04298-f011:**
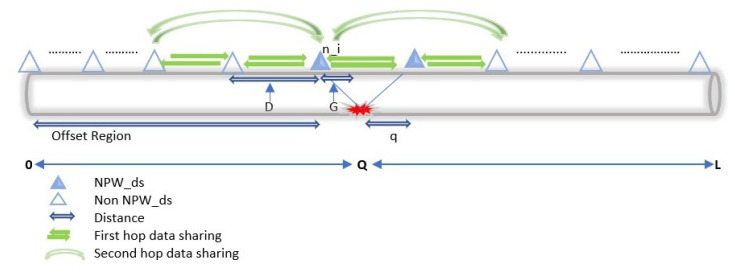
Detection and localisation of leakages.

**Figure 12 sensors-23-04298-f012:**
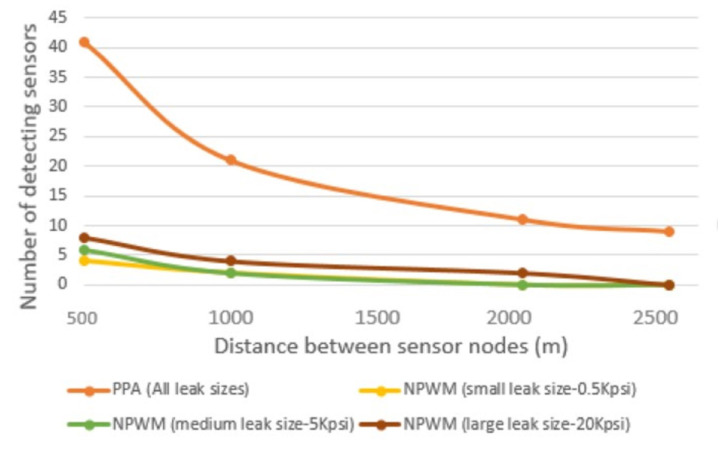
Detectability of leakages by the number of sensors and distance.

**Figure 13 sensors-23-04298-f013:**
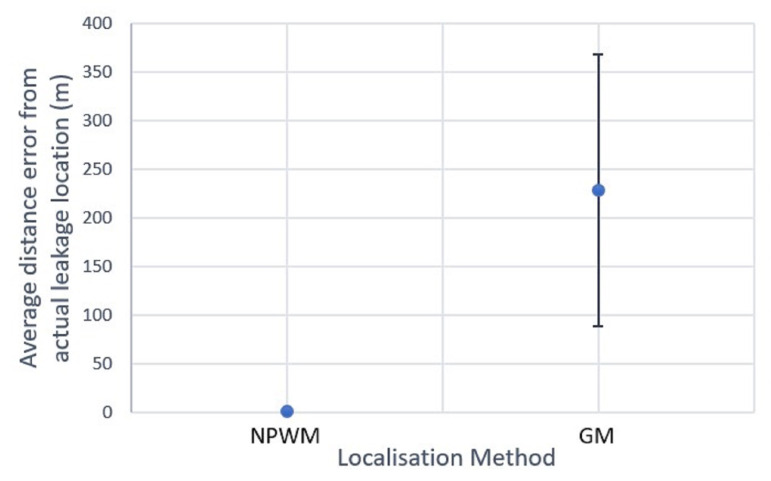
Average localisation accuracy of classical NPWM and GM.

**Figure 14 sensors-23-04298-f014:**
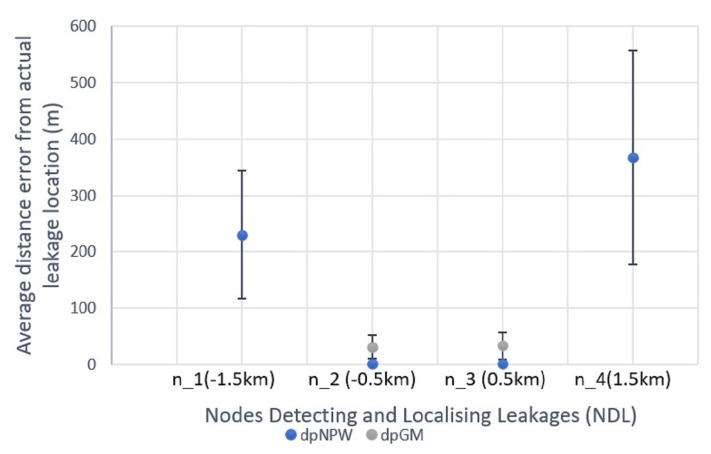
HyDiLLEch-1 average localisation accuracy by NDLs.

**Figure 15 sensors-23-04298-f015:**
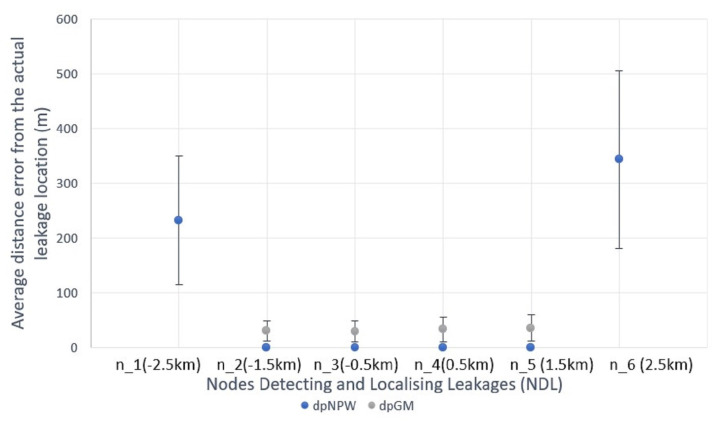
HyDiLLEch-2 average localisation accuracy by NDL.

**Figure 16 sensors-23-04298-f016:**
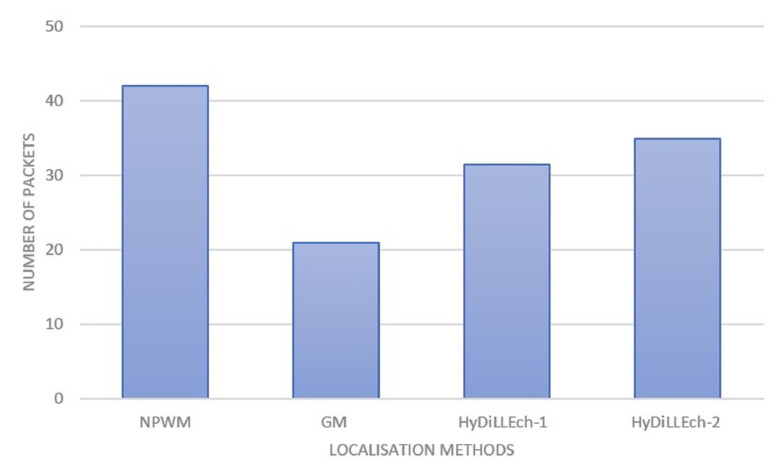
Communication overhead by the number of packets.

**Figure 17 sensors-23-04298-f017:**
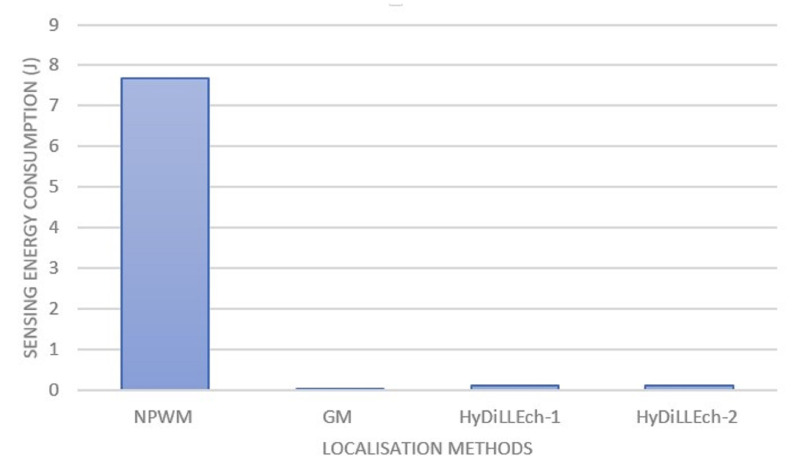
Sampling energy consumption of the sensors.

**Figure 18 sensors-23-04298-f018:**
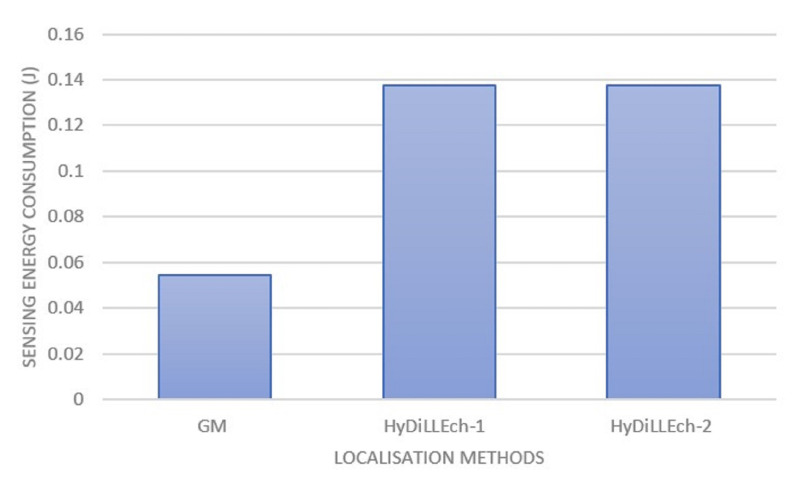
Sampling energy consumption of the LDTs with separated DAL phases.

**Figure 19 sensors-23-04298-f019:**
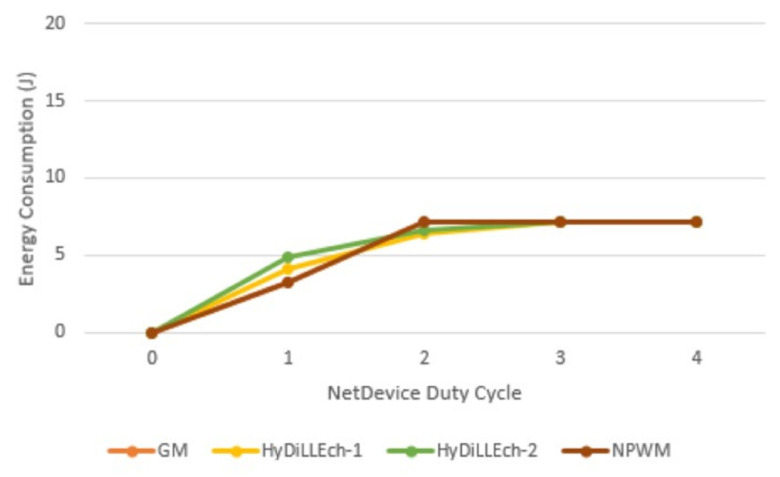
Radio energy consumption.

**Figure 20 sensors-23-04298-f020:**
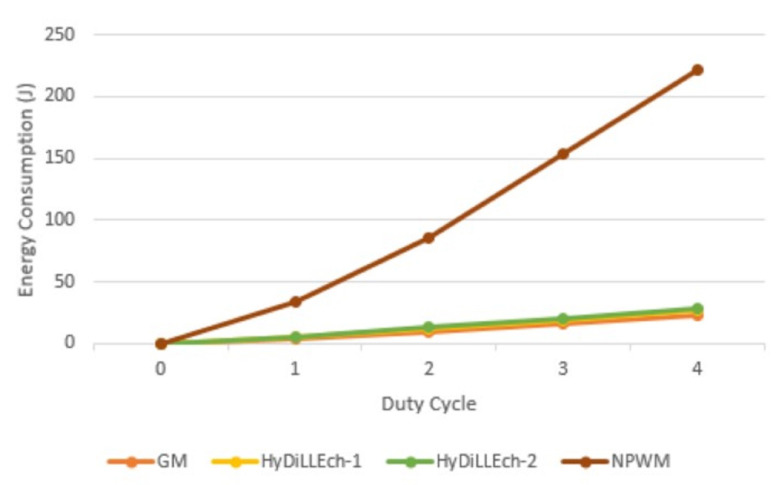
Cumulative energy consumption.

**Table 1 sensors-23-04298-t001:** Pipeline and oil characteristics.

Pipeline/Oil Characteristics	Value/Type
Material	Carbon Steel
Pipeline Length (*L*)	20 km
Wall thickness (*w*)	0.323 m
Inside diameter (*d*)	0.61 m
Height/elevation (*z*)	0 m
Oil kinetic viscosity	2.90 mm${}^{2}$/s
Temperature	50 ${}^{\circ }$C
Oil density (ρ)	837 kg/m${}^{3}$
Inlet pressure ($P_{0}$)	1000 psi
Reynolds no ($R_{e}$)	1950
Velocity (*V*)	2 m/s
Molecular Mass (*m*)	229
Oil elasticity (*K*)	$1.85\times 10^{5}$ psi
Carbon steel elasticity (*Y*)	$3\times 10^{6}$ psi
Gravitational force (*g*)	9.81 m/s${}^{2}$
Constant (*e*)	2.718
Coefficient of friction (λ)	0.033
Wave speed (*c*)	14.1 m/s

**Table 2 sensors-23-04298-t002:** Network simulation parameters.

Network Parameters	Value/Type
Number of sensors	21
Number of gateways	1
PHY/MAC model	802.11ax Ad hoc
Transmit power	80 dBm
Transmit distance	20 km
Error model	YANS
Propagation Loss	Log-distance
Path Loss ($L_{0}$)	46.67 dB
Reference distance ($d_{0}$)	1 m
Path-Loss Exponent (σ)	3.0
Packet size	32 bytes
Data rate	1 Kbps
Distance between sensors	1 km
Duty cycle	70%
Low Frequency	s
High Frequency	ms

## Data Availability

Restrictions apply to the availability of these data. Data was obtained from the Department of Petroleum Resources with a non-disclosure agreement and are not publicly available.

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
