# Peer review of "Development and Analysis of a Distributed Leak Detection and Localisation System for Crude Oil Pipelines"

_sensors, 2023, doi:10.3390/s23094298_

Round 1

Reviewer 1 Report

The authors have conducted interesting work. The detection of leakages in crude oil pipelines.

Author Response

Dear Sir/Ma,

I would like to thank you for your valuable feedback. We will review the article to correct all spelling mistakes and other grammatical errors before resubmission.

Best regards, 

Safuriyawu Ahmed

Reviewer 2 Report

1.      The references lack uniformity and some of them needs correction.

2.      The paper has some grammatical errors. Check thoroughly for errors once again.

3.      The conclusion needs more explanation.

4.      Write the future work in a little more detail. Also mention the limitation of the paper.

5.      Use different colours for figure 12. All of them are similar and does not provide clarity.

6.      Figure 15 lacks clarity as compared to figures 13 and 14.

7.      Use brighter colours for figures 19 and 20. Apart from purple, others look dull and in figure 20 it is hard to distinguish yellow and blue lines.

8.      Re-check the heading formats once again throughout the paper.

9.      There is a lot of unnecessary space between figures 18 and 19.

10.  There are many abbreviations in the paper. Either mention them in a similar format or mention them in a tabular format for better understanding.

11.  The citations need to be corrected in the paper. For example, there are many citations written as “Berry et al. in their work [31]”, which should be written as “Berry et al. [31] in their work”.

Author Response

Dear Sir/Ma,

Thank you very much for your valuable feedbacks. Kindly find below, a point-by-point response to each feedback you provided:

  1. I have checked the references. Some of the DOIs are incorrect and are automatically pulled by the editing package. I will discuss this with the editors.
  2. I have gone through the entire article to correct all grammatical errors.
  3. I added more explanations to the conclusion.
  4. I also added more explanations to the future works.
  5. Done
  6. Done
  7. Done
  8. Done
  9. I used the vertical space function to reduce the space but it does not have effect. Latex automatically set the space between the figures.
  10. A list of acronyms has been included in the article.
  11. Done.

Once again, thank you.

Best regards,

Safuriyawu Ahmed

Reviewer 3 Report

The paper presents a Hybrid Distributed Leak Detection and Location (HyDiLLEch) technique comprising multiple classical LDTs and implemented in two versions with which higher sensitivity to leak detection and location and elimination of SPOF related to centralized LDTs is achieved and leak accuracy is improved at simulation level.

The document is well presented and the method is appropriately defined. The systematic review of traditional techniques can be highlighted. The development of a solution to the problem posed as the main objective of the research is clearly achieved by showing the advantages of the proposed method. It is recommended to take the method to a real system or prototype to verify its effectiveness in a real environment.

I consider that the keywords are not appropriate to give visibility to the document so I recommend to revise and expand

Author Response

Dear Sir/Ma,

Thank you very much for your valuable feedbacks.

We have included experimental validations of the work as part of future works. We also added more keywords as you have suggested.

Best regards,

Safuriyawu Ahmed

Reviewer 4 Report

This paper is a continuation of the recent work of author’s lab, and designs a Hybrid and Distributed Leakage detection and Localisation technique (HyDiLLEch) comprising multiple classical LDTs and implemented in two versions (single and double-hop) for a distributed leakage detection and localisation system for crude oil pipelines, which is presented in Section 4.

The simulation results in Section 5 show that:

1. The optimal distance between the sensors is experimentally determined as 1000 m to enable sensitivity to small leakages;

2. HyDiLLEch successfully eliminates SPOF associated with the centralised systems with the increment of NDLs from one to four and six for both the single and double-hop versions of the algorithm;

3. The accuracy of DAL in nodes closest to the leak point is improved to an average of 15 metres (0 metres to 32 metres) with minimised energy consumption and communication overhead.

Overall, I find this paper clearly written. The idea is interesting and the obtained result is valuable. I recommend this paper for publication in this journal.

Author Response

Dear Reviewer, 

Thank you very much for your valuable feedbacks.

You have recommended extensive editing of the English Language and writing style and we would like to work on that aspect before resubmission.

Could you kindly provide us some further details regarding this recommendation and how we can improve the article.

Please accept the assurance of our highest regards as we look forward to your response.

Yours Sincerely,

Safuriyawu Ahmed.

Reviewer 5 Report

In this study, the authors developed and analyzed a distributed leak detection and localisation system for crude oil pipelines. I have the following suggestion for improving this manuscript:

1- The abstract did not mention the study methodology.

2- Line 9: Please briefly state the results that led to the increased sensitivity to leakage detection ... etc.

3- Please add study objectives before contributions details.

4- Please add more related studies and the novelty of your research compared with the previous studies.

5. The discussion section needs improvement, please compare the current study with previous studies.

6. I think that the authors should mention the outputs and recommendations of the study and the limitations of the study in the conclusions section.

Author Response

Dear Sir/Ma,

Thank you very much for your valuable feedbacks. Kindly find below, a point-by-point response to your reviews:

  1. Study methodology has been added to the abstract.
  2. This has also been added to the abstract.
  3. The objectives have been moved to the beginning of the contributions section.
  4. This has also been added.
  5. For the results presented, both NPWM and GM represent the classical approaches with which we carried out the comparative analysis. The analysis are in terms of resilience to single point of failure, the accuracy of detection and localisation, and the corresponding communication overhead for each technique.
  6. The concluding section has also been revised.

Thank you once again. We are at your your disposal should you have further recommendations.

Best regards,

Safuriyawu Ahmed.

Reviewer 6 Report

According to the main content, there are several weaknesses as follows:

1- There is no novelty in the evaluation algorithm.

2- What is exactly main novelty to illustrate node placement strategy for distributed detection and localisation of leakages ?

3- The main contributions of this research are unclear. For example, why authors  Combined existing LDTs in a distributed manner ??

4- All shapes and figures (Specially, Figures 7, 8, 9, 10, and 11) are unclear. There is no a technical aspect in each figure. reader just can see some shapes and lines.

5- Time complexity of the proposed HyDiLLEch (Double-Hop) should be evaluated.

6- Existing initial parameters for simulation in a WSN environment is unclear.

Author Response

Dear Sir/Ma,

Thank you very much for your valuable feedbacks. Kindly find below the response to the reviews you provided:

  • We have given further explanations to the novelty of the work in the introduction section and also the contribution section.
  • Figures 7,8,9,10 and 11 were intended to give a visual representation and explanation of the the system.
  • We are working on evaluating the time complexity of both versions and we have included it as part of future works.
  • We have also spell-checked the entire document. 

Thank you once again. We are at your disposal should you have further recommendations.

Best regards,

Safuriyawu Ahmed

Round 2

Reviewer 2 Report

Can be accept now

Author Response

Dear Sir/Ma,

We thank you for taking out time to review our work.

Best regards,

Safuriyawu Ahmed

Reviewer 5 Report

The authors responded to my suggestions, and I think the manuscript has improved significantly.

Author Response

Dear Sir/Ma,

We would like to thank you for taking out time to review our work. 

Best regards,

Safuriyawu Ahmed

Reviewer 6 Report

Authors have addressed technical comments in this revisision.

It is recommended that authors can discuss on time complexity of the Algorithm 2 as HyDiLLEch (Double-Hop).

Author Response

Dear Reviewer,

We would like to thank you very much for your valuable feedback. The analysis of the time complexity has now been added to the article.

Best regards,

Safuriyawu Ahmed

Round 3

Reviewer 6 Report

Authors have addressed all my comments in the main text